# OpenReview forum: "Don’t fear the unlabelled: safe semi-supervised learning via debiasing"
_ICLR.cc/2023/Conference — ICLR 2023 poster_

### Official Review · Reviewer_DVba · 2022-10-23

**Confidence:** 3
**Clarity, Quality, Novelty And Reproducibility:** The paper is clearly structured and w…
**Correctness:** 4
**Technical Novelty And Significance:** 3
**Empirical Novelty And Significance:** 3
**Recommendation:** 6

**Strength And Weaknesses:**

Strength:
* A simple debiasing method with strong and comprehensive theoretical backing.

Weakness:

I am rather happy with the simplicity of the method and the theoretical backing of the approach, and just have some suggestions in terms of experiment evaluation, and in terms of adding theoretical comments around assumption violation.

* Missing intrinsic evaluation: Authors derived variance-reduction and calibration property of the method. Consequently, I do expect to see some experiment results on risk estimation quality (bias and variance) and calibration property (e.g., measured by proper scoring rule) to validate the theoretical argument in practice.
* Missing theoretical comments / empirical discussion on how the method may perform if MCAR is violated.
* Investigation is restricted to simple academic benchmarks, evaluation on realistic large-scale benchmarks (e.g., ImageNet) with rich data distribution and large number of output categories is missing. (Although I believe the current empirical evaluation is sufficient given the theoretical contribution of the paper).

**Summary Of The Paper:**

This work proposes DESSL, a general methodology to debias the risk estimate of the SSL objective under the MCAR assumption (Equation 5). Authors conducted comprehensive theoretical investigation of the approach, including its connection with control variates and constrained optimization, its finite-sample guanrantee in estimation variance, its asymptotic guanrantee in calibration, consistency and normality, as well as its generalization bound in terms of Radamacher complexity (Section 3). Authors applied DESSL to popular SSL methods (EntMin, PseudoLabel and FixMatch) and tested them on simple benchmarks (MNIST, CIFAR, MedMNIST), showing clear improvement over baselines.

**Summary Of The Review:**

This work proposes a simple approach to improve risk objective estimation in SSL. The method is simple to implement, backed by comprehensive theory, and is sufficiently evaluated on standard benchmarks. I believe the experiments should be extended to very some of the key theorems in practice (e.g., variance reduction and calibration), but otherwise the quality is sufficient to pass the ICLR bar.

---

> ### Author Response · Authors · 2022-11-11
> **Answer to reviewer DVba**
>
> Many thanks for your comments and assessment of our paper!
>
> >I do expect to see some experiment results on risk estimation quality (bias and variance) and calibration property (e.g., measured by proper scoring rule) to validate the theoretical argument in practice.
>
> Thank you for this suggestion. To assess the debiasing and the variance reduction properties, we trained a CNN-13 using 4,000 labelled data. We then estimate the value of the risk computed on the 10,000 test points using only 40 labelled data points and 9,960 unlabelled data points and compute the DePL and its biased counterparts. The experiments show that DePL is an unbiased estimator of the risk for any value of $\lambda$ and its variance is lower than the variance of the Complete Case of $\lambda$ close to $\lambda_{opt}$.
>
> Concerning the calibration, we always evaluate our model using the log-likelihood which is a proper scoring rule. Our results show that DeSSL is better calibrated than SSL and the Complete Case (see Table 1). Also, the toy example in Figure 1 illustrates that DeSSL is as well calibrated as the Complete Case (as stated by Theorem G.1) whereas its SSL counterparts are not. However, we added the Brier score in our experiments on Fixmatch.
>
> >Missing theoretical comments / empirical discussion on how the method may perform if MCAR is violated.
>
> MCAR means $r$ (the indicator of being labelled) is independent of $x$ and $y$. This property is important for all our theoretical results to simplify the expressions of expectation and variance. Indeed, we need the assumption to prove that DeSSL is an unbiased estimator of the risk. Also, we remark that without this assumption, the complete case (using only labelled data) is not an unbiased estimator of the risk (see e.g. Liu and Goldberg, 2020, Section 3.1). Since our definition of safety is tied to the performance of the complete case, in non-MCAR settings, it is not straightforward to extend it.
> For these reasons, it is difficult to go beyond the MCAR assumption but maybe we can find an application in MAR settings ($r{\perp}y|x$). For the particular case of the likelihood, the complete case in a MAR setting has all the good properties of maximum likelihood (see Chapelle et al., 2006, section 9.2.1). In regards to Equation 6, our method can be expressed as a maximisation of the likelihood under a constraint. Therefore, our method will work if we make the assumption on the data that the quantity $H$ is on average equal on labelled data and unlabelled data (i.e. $\mathbb{E}[H|r=1]= \mathbb{E}[H|r=0]$). Also, the work of Liu and Goldberg makes us conjecture that a variant of our method can be derived on a MAR setting as they also use an unbiased estimate of risk, like ours, for a specific choice of $H$. We added a short explanation of how important is the MCAR assumption for our definition of safeness in section 2.2.
>
> >Evaluation on realistic large-scale benchmarks (e.g., ImageNet) [...] is missing.
>
> Unfortunately, as presented in Appendix O, SSL methods require a lot of computing resources and time to be trained. For instance, Fixmatch's authors said about ImageNet in the issue \#31 of the official GitHub repository "We trained on TPU with 32 cores, which should be roughly equivalent in terms of compute to 32 v100. [...] In our setup it took about 2.5 - 3 days to train for full 3000 epochs.".

---

> > ### Comment · Reviewer_DVba · 2022-11-18
> > **Thanks for the updates!**
> >
> > Thanks authors for the additional experiments to evaluate the theoretical arguments. I believe my comments regarding missing intrinsic evaluation and theoretical comments are reasonably addressed.
> >
> > A minor follow-up point (as this has surfaced in some other reviewers' comments): since the name "safe SSL" can carry alternative definitions (e.g., as that pointed out by [6iPj](https://openreview.net/forum?id=TN9gQ4x0Ep3&noteId=t2FAwx7HQc)). Maybe to avoid confusing readers and more to appropriately highlight the core contribution of the work, it is helpful to update the core positioning of the paper from "safe SSL" to something like "unbiased SSL"? I wonder do the authors think? Thanks.

---

> > > ### Author Response · Authors · 2022-11-18
> > > **Answer to reviewer DVba**
> > >
> > > Thanks for engaging with us, reading other reviews, and suggesting to replace "safe" with "unbiased". Indeed, the name "safe SSL" has several meanings, and is more a general branch of research than a precise mathematical concept.
> > >
> > > However, we do believe that our work is part of this branch, since we share with the rest of the "safe SSL" literature the common goal of getting guarantees about being as good as the complete case. Even if these definitions are slightly different, they all go in the same direction. The sort of properties that we get is also (qualitatively at least) similar to the ones obtained by other "safe SSL" papers. For instance, Mey and Loog [2019] review several contributions that derive asymptotic normality of SSL estimates, or Rademacher generalisation bounds.
> > >
> > > We also note that *unbiased* SSL is another branch of SSL where labelled and unlabelled data do not have the same distribution (see Fox-Roberts and Rosten [JMLR 2014]) or class-imbalance settings (see Liu et al. [ICLR 2021] and Wang et al. [CVPR 2022]).
> > >
> > > For these reasons, we chose to keep the reference to "safe SSL".
> > >
> > > Additional references:
> > >
> > > [1] Liu et al., *Unbiased Teacher for Semi-Supervised Object Detection*, ICLR 2021
> > >
> > > [2] Wang et al., *Debiased Learning From Naturally Imbalanced Pseudo-Labels*, CVPR 2022

---

### Official Review · Reviewer_6iPj · 2022-10-24

**Confidence:** 5
**Correctness:** 2
**Technical Novelty And Significance:** 2
**Empirical Novelty And Significance:** Not applicable
**Recommendation:** 6

**Clarity, Quality, Novelty And Reproducibility:**

The proposal is simple and easy to follow. But the theoretical and experimental parts are not convincing.

**Strength And Weaknesses:**

Strength:
1) The proposal is clear and easy to understand.
2) The proposed debias method is general and can be applied to various semi-supervised learning methods.

Weakness:
1) The safety of semi-supervised learning means its performance will be better than simple supervised learning. Why the deep SSL, such as FixMatch will be unsafe? From the experimental results of other SSL methods, it seems they do not suffer from this problem.
2) The theoretical analysis seems did not give the guarantee that the performance of the proposed method will be better than the supervised learning method. So it may be inconsistent with the author's claim.
3) The experimental results are not convincing. The author should apply the proposal to more SSL methods (such as PL, MixMatch, FixMatch, ReMixMatch, UDA, etc) and more commonly adopted datasets (such as STL-10, SVHN, Image-Net).
4) The paper writing needs to be improved. For example, Figure 1 needs to be reorganized.

**Summary Of The Paper:**

This paper tries to give theoretical analysis about the safeness of semi-supervised learning, and proposed a debiased method that can be applied to the classical semi-supervised learning objectives. Experimental results show the proposal can improve the performance of FixMatch algorithm.

**Summary Of The Review:**

Based on the above discussion, the paper is not ready to be published, so I tend to reject this paper.
--------------------------------

The author's responses address my concerns. Based on the author's efforts and other reviewers' comments,  I change the score to borderline accept.

---

> ### Author Response · Authors · 2022-11-11
> **Answer to reviewer 6iPj**
>
> Many thanks for your comments and assessment of our paper!
>
> >The safety of semi-supervised learning means its performance will be better than simple supervised learning.
>
> We define in the introduction of the paper the safeness of an SSL method: "*an SSL algorithm is safe if it has theoretical guarantees that are similar or stronger to the complete case baseline*". It is unfortunately impossible to ensure performance improvement without strong assumptions on the data distribution. We discuss this point in the first paragraph of part 2.3 in our paper and Appendix C. For instance, S4VM (Li et al., 2014) ensures better performance using the low-density assumption and considering having access to the true model. On the other hand, Schölkopf et al. (2012) show that depending on the causal relationship between the data and the labels, SSL may always fail. Therefore, ensuring stronger or similar theoretical guarantees is an adequate definition of safe for SSL.
>
>
> >Why the deep SSL, such as FixMatch will be unsafe?
>
> Indeed, the overall accuracy of the classic SSL experimental setup does not show the limitation of Fixmatch. First, recently, several papers show the overall accuracy hides the disparate effect of SSL on sub-populations. Indeed, in Zhu et al. (ICLR 2014), the authors the classic methods such as MixMatch and UDA benefit the subpopulations with a higher complete case accuracy (the "rich"). Also, Chen et al. (2022, arXiv:2202.07136) show that the poor classes have a performance drop by introducing Fixmatch compared to their Complete case accuracy. Secondly, as highlighted in several works, SSL may fail in real-world applications where datasets are less favourable (Singh et al., 2008; Schölkopf et al., 2012; Li & Zhou, 2014).
>
> >The theoretical analysis seems did not give the guarantee that the performance of the proposed method will be better than the supervised learning method.
>
> As explained earlier, it is particularly difficult to guarantee better performances without strong assumptions on the data distribution. However, we prove in Theorem 3.5 that DeSSL is asymptotically normal and has a better asymptotic variance than the complete case for $\lambda$ in $[0, 2\lambda_{opt}]$ with mild assumptions on the loss and the data distribution.
>
> >The experimental results are not convincing. The author should apply the proposal to more SSL methods (such as PL, MixMatch, FixMatch, ReMixMatch, UDA, etc) and more commonly adopted datasets (such as STL-10, SVHN, Image-Net).
>
> We decided to compare our method to Fixmatch as the current state-of-the-art method and similar to the other methods (MixMatch, ReMixMatch and UDA). We did compare ourselves to PseudoLabel in Figures 2 and 3. Unfortunately, as presented in Appendix O, SSL methods require a lot of computing resources and time to be trained. For instance, Fixmatch's authors said about ImageNet in the issue \#31 of the official GitHub repository "We trained on TPU with 32 cores, which should be roughly equivalent in terms of compute to 32 v100. [...] In our setup it took about 2.5 - 3 days to train for full 3000 epochs.".
>
> >The paper writing needs to be improved. For example, Figure 1 needs to be reorganized.
>
> Can you elaborate on your concerns regarding the organisation of Figure 1 and the writing of the paper? We would benefit a lot from your comments.

---

### Official Review · Reviewer_1cgk · 2022-10-27

**Confidence:** 4
**Correctness:** 3
**Technical Novelty And Significance:** 3
**Empirical Novelty And Significance:** 3
**Recommendation:** 8

**Clarity, Quality, Novelty And Reproducibility:**

The paper is well written, the structure is well organized and the content is easy to follow. The debiasing idea in the context of Semi Supervised learning (SSL) is presented clearly with minimum fuzziness.

There have been a number of innovations especially the debiasing idea itself plus its theoretical derivation including proofs of some of the underlying theorems.  Another advantage is the DeSSL method can easily be applied to some  existing SSL algorithms without making any assumptions on the distribution of data.  The paper also provided some theoretical guarantee estimates  (on the safeness of the proposed methodology) on consistency, calibration, asymptotic normality and generalization error.

As the code has been shared, it is expected the results can be reproduced.


**Strength And Weaknesses:**

Strength: The proposed DeSSL model is based on the hypothesis that existing SSL techniques put less weight on unlabeled data and more on the labelled set and thus bias the learning. If this biasing problem can be solved, it is likely that SSL model performance will improve. The debiasing methodology, presented in this work, comes with a set of theorems and corresponding proofs that make the paper theoretically strong. The experiment results, especially the boosted worst-class  performance, signals that  the debiasing idea is working to an extent (for classification tasks).

The proposed approach has been tested on CIFAR-10, CIFAR-100 and compared against FixMatch, a known technique in SSL.  The most interesting result is the DeSSL worst-case accuracy, which is noticeably better than FixMatch for both CIFAR-10 and 100 benchmarks although the overall accuracy gain for CIFAR-100 was marginal.

An additional advantage is that the proposed  DeSSL method can be easily applied to some of the available SSL algorithms with minimal changes/efforts.

Weakness: Although the results reported in section 4 look encouraging it is difficult to judge whether the reported gains are statistically significant. It is suggested that authors perform some statistical tests to measure if the performance gains are statistically significant.

The overall accuracy gain for CIFAR-100 was marginal (compared to CIFAR-10) which signals scalability limitations of the approach.


**Summary Of The Paper:**

Empirical Risk Minimization (ERM) is an established technique and is one of the basis of supervised (plus some other types) learning. The ERM objective is to minimize the risk when fitting a set of labeled data points. In the context of Semi Supervised Learning (SSL), as we need to deal with data points with and without labels, existing conventional approaches combine (usually through some weighting) two independent losses; however, they lack concrete theoretical guarantee (one of the claims of this paper). One of the the objectives of this paper is to address this issue and ensure the unbiased (in the context of data distribution) semi supervised learning.

Traditional SSL techniques (such as low density separation and consistency regularization based) heavily depend on labeled data points and sometimes overweight them when compared to the corresponding unlabeled set. This may look obvious as unlabeled data come with label uncertainty (compared to labelled data). The hypothesis is even unlabeled data come with uncertain labels; they shouldn’t be weighted poorly (compared to labeled data) to learn a proper model.

In this work (DeSSL), the SSL problem has been formulated as s ERM problem which also includes a debiasing term to control/minimize the labeled data bias. The main hypothesis is unlabeled data should be taken as complementary not the competitor of the available labeled data. The debiasing term can be thought as a regularizer to penalize higher weights to labeled data and thus control the data bias.

The proposed approach has been tested on CIFAR-10, CIFAR-100 and compared against FixMatch. Reported results are found to be encouraging.

**Summary Of The Review:**

I have gone through the paper more than once including the appendices. Overall, the debiasing idea is quite sound, well articulated, and the document is found easy to follow. Learning by balancing the contribution from both labeled and unlabelled data is well supported through experiments and  corresponding results. Some important SSL theoretical derivations have made the paper even stronger.

---

> ### Author Response · Authors · 2022-11-11
> **Answer to reviewer 1cgk**
>
> Many thanks for your comments and assessment of our paper!
>
> >It is suggested that authors perform some statistical tests to measure if the performance gains are statistically significant.
>
> We perform statistical tests on our results with DePseudoLabel on CIFAR-10 in Appendix M. These tests show that DePseudoLabel is significantly better than PseudoLabel for $\lambda$ close to 10. We will add a statistical test for our results with Fixmatch on CIFAR-10. Thank you for this suggestion.
>
> >The overall accuracy gain for CIFAR-100 was marginal (compared to CIFAR-10) which signals scalability limitations of the approach.
>
> On CIFAR-100 (respectively CIFAR-10), DeFixmatch improves Fixmatch overall accuracy by 1.94% (resp. 1.57%) and the Complete Case overall accuracy by 8.60% (resp. 8.17%).

---

### Official Review · Reviewer_NKdq · 2022-10-28

**Confidence:** 3
**Correctness:** 4
**Technical Novelty And Significance:** 4
**Empirical Novelty And Significance:** 2
**Recommendation:** 8

**Clarity, Quality, Novelty And Reproducibility:**

The paper is well written and clear. They have included the relevant prior work. The idea of computing the entropy term on the labelled data as well to make the loss estimator unbiased is novel in this context to the best of my knowledge.

They have also released the code for their experiments.

**Strength And Weaknesses:**

Semi-supervised learning is an interesting area and coming up with theoretically sounds methods is an important problem. The idea of debasing the risk is simple and interesting and also leads to an unbiased estimator of the risk with theoretical guarantees. The toy example provided is appealing. Their method also leads to improved calibration and better accuracy on certain subgroups.

The main weakness is that their method does not lead to improved accuracy on many of the datasets. It would be nice to have some discussion of the properties of the dataset which leads this method to outperform others in terms of accuracy.  They also show that the debiased version of Fixmatch leads to better accuracy but not the debased version of Pseudolabels. Is there some reason for why this is the case? It would also be good to discuss why their method leads to better accuracy on certain subgroups when using Fixmatch.

**Summary Of The Paper:**

This work identifies that the existing approaches in semi-supervised learning minimize a biased risk and hence, are devoid of theoretical guarantees unless there are strong assumptions. This works gives a method to de-bias the loss with a simple estimator which also allows them to give theoretical guarantees on the risk. They also evaluate their debiased versions of existing semi supervised learning approaches on various datasets and show that they lead to improved calibration and improved accuracy in certain settings.

**Summary Of The Review:**

The idea is interesting, simple and natural with theoretical guarantees. The paper is very well written. But, I think the experimental section is weak as discussed above.

---

> ### Author Response · Authors · 2022-11-11
> **Answer to reviewer NKdq**
>
> Many thanks for your comments and assessment of our paper!
>
> >It would be nice to have some discussion of the properties of the dataset which leads this method to outperform others in terms of accuracy.
>
> Our experiments suggest that DeSSL will perform as well as SSL when the classic SSL assumptions are true (such as the cluster assumption), for instance with MNIST in Figure 2. Still, for MNIST, DeSSL does not outperform in terms of accuracy but does in terms of cross-entropy and expected calibration error (see Figure 7). However, when these assumptions do not hold anymore DeSSL will outperform their SSL counterparts as you can see with CIFAR-10 and CIFAR-100 (Table 1) or the toy example (Figure 1). However, DeSSL does not degrade SSL performances as SSL may do for instance with dermaMNIST (Figure 4) when the classes are unbalanced. This is an interesting discussion that has added to section 5 (conclusion).
>
>
> >They also show that the debiased version of Fixmatch leads to better accuracy but not the debiased version of Pseudolabels. Is there some reason for why this is the case?
>
> Indeed, DePseudoLabel does not outperform (in terms of accuracy) PseudoLabel on MNIST but does on CIFAR-10 for $\lambda$ close to 10, and DermaMNIST. As shown in Theorems 3.1 and 3.5, DeSSL has variance reduction properties in $[0,2\lambda_{opt}]$ thus we expect DeSSL to outperform SSL close to $\lambda_{opt}$, and not necessarily for all $\lambda$.
>
> >It would also be good to discuss why their method leads to better accuracy on certain subgroups when using Fixmatch.
>
> Thank you for this suggestion, we will add this discussion in section 4.2. Indeed, as explained by Zhu et al. (ICLR 2022), pseudo-label-based methods like Fixmatch will benefit the subpopulations with a higher accuracy baseline ("easy" subpopulations). On the other subpopulations, we can even observe a drop in performance by adding SSL (see Figure 4). Indeed,  pseudo-label-based methods with a fixed selection threshold draw more pseudo-labels in the "easy" subpopulations. For instance, in the toy dataset, PseudoLabel will always draw samples of the class blue in the overlapping area. DeSSL prevents the method to overfit the "easy" classes with the debiasing term and then being overconfident on the same "easy" subpopulations. Recently, several methods try to fix this drawing problem with an adaptative threshold such as Zhang et al. (NeurIPS 2021).

---

### Author Response · Authors · 2022-11-11
**General comments**


We thank the reviewers for their valuable feedback. All the reviewers agreed that our "paper is well written and clear" (**Reviewer NKdq**), "the proposal is clear and easy to understand" (**Reviewer 6iPj**), "the content is easy to follow" (**Reviewer 1cgk**), or that "the paper is clearly structured and well written" (**Reviewer DVba**). We particularly appreciate their assessment that the debiasing method is "simple and interesting" (**Reviewer NKdq**) and comes with a "strong and comprehensive theoretical backing" (**Reviewer DVba**). They note that on the more practical side, the “method can be easily applied to some of the available SSL algorithms with minimal changes/efforts.” (**Reviewer 1cgk**) and that "the method is general and can be applied to various semi-supervised learning methods" (**Reviewer 6iPj**). Reviewers also assessed the novelty of the method: "The idea [...] is novel" (**Reviewer NKdq**) and "There have been a number of innovations" (**Reviewer 1cgk**).

Following the reviewer's remarks, we have made the following modifications to the paper (modifications are in red in the PDF):

* As suggested by **Reviewer 1cgk**, we performed a paired Student’s t-test to ensure that our results are significant in both accuracy, cross-entropy and Brier score for Fixmatch on CIFAR-10 and reported the p-values in section Appendix N.
* As suggested by **Reviewer DVba**, we added an experiment to validate the quality of the estimation of the risk and its gradient in practice. We trained a CNN on CIFAR-10 using only 4,000 labelled data and then used the test dataset to evaluate the PseudoLabel and DePseudoLabel risks that we compared to the compare to the complete case risk using all the test set (see Appendix E.1) to illustrate the unbiasedness of DeSSL. We also evaluated the variance of the DePseudoLabel risk with varying the hyperparameters $\lambda$ to illustrate the variance reduction theorem. Additionally, we showed that DePL considerably reduces the variance of the gradient of the risk which is essential when optimising with gradient methods.
* As suggested by **Reviewer DVba**, we added the Brier score in our experimental results on Fixmatch to validate the calibration result of Theorem G.1 as the Brier score is a proper scoring rule. These results show that DeFixmatch is better calibrated than Fixmatch and the Complete Case.
* As suggested by **Reviewer DVba**, we added a short explanation of how important is the MCAR assumption for our definition of safeness in section 2.2.

We thank you again for the good quality of your comments and remarks and look forward to any additional questions or suggestions.

---

### Author Response · Authors · 2022-11-18
**General comment**

**Reviewer 1cgk**
>The overall accuracy gain for CIFAR-100 was marginal (compared to CIFAR-10) which signals scalability limitations of the approach.

**Reviewer 6iPj**
>The author should apply the proposal to [...] more commonly adopted datasets (such as STL-10, SVHN, Image-Net).

**Reviewer DVba**
>Investigation is restricted to simple academic benchmarks, evaluation on realistic large-scale benchmarks (e.g., ImageNet)

As explained in the individual answers, SSL methods require a lot of computing resources and time to be trained, in particular on the Imagenet dataset. To address these scalability issues, we have just added an experiment on STL-10 that complements the ones on Cifar10/100 The STL-10 dataset is a subset of the Imagenet dataset that contains 5,000 labelled images from 10 classes and 100,000 unlabelled images. We trained a WideResnet37-2 using 4,200 labelled data for training and 800 for validation. We report the accuracy, worst class accuracy, cross-entropy and brier score below and in section 4.2 of the pdf. On STL10, the performance gain are smaller than on CIFAR10 and CIFAR100. It would be interesting to compare these results with a fully supervised method. However, DeFixmatch outperforms both Fixmatch and the Complete Case in all metrics reported.

| STL-10               | Complete Case | Fixmatch | DeFixmatch |
|:-------------------- |:-------------:|:--------:|:----------:|
| Accuracy             |     88.69     |  92.64   | **93.23**  |
| Worst class accuracy |     79.75     |  85.00   | **85.25**  |
| Cross-entropy        |     0.42      |   0.30   |  **0.28**  |
| Brier score          |     0.171     |  0.117   | **0.109**  |

---

### Decision · Program_Chairs · 2023-01-20

**Decision:**

Accept: poster

**Justification For Why Not Higher Score:**

The paper presents several interesting theoretical conclusions derived from a novel SSL risk objective, which can potentially lead to significant real-world impact.  However, the current empirical results, while sufficient for publication, I think are not conclusive enough to understand the broader practical impact of the work.

**Justification For Why Not Lower Score:**

The paper presents a novel approach with theoretical insight as well as some initial empirical motivation -- it it deserves to be presented at this venue.

**Metareview: Summary, Strengths And Weaknesses:**

This work presents a novel SSL training objective, which specifically addresses the issue of bias that can be introduced when using other standard SSL approaches.  The proposed SSL approach is theoretically motivated by proving the consistency of the empirical proposed risk and also by providing a finite sample generalization bound. Both of these are proven under certain assumptions, such as labels "missing completely at random" (MCAR).  The approach is further validated empirically using benchmarks datasets such as MNIST, CIFAR-10/100 and compared the original and unbiased variants of various SSL approaches (e.g. Fixmatch vs. DeFixmatch, PseudoLabel vs DePseudoLabel). In cases where certain additional assumptions hold (such a clusterability), the authors find similar performance to existing biased approaches, but in other cases find improvement over biased counterparts, especially in sub-metrics such as worst class accuracy.

Reviewers all agree that additional empirical evaluations can further strengthen the paper (and I acknowledge that the authors have already provided additional empirical results since the discussion), but they also agree that the stronger contributions of the paper are the theoretical conclusions. Taken together, I recommend accepting the paper for publication.

One final request to the authors is to clarify and caveat the use of the phrase "safe SSL" distinguishing it from its use in other lines of SSL work (which has already been done to some extent in the revised submission), as this was one strong point of contention during discussion.


**Note From Pc:**

if the above contains the word "oral" or "spotlight" please see: "oral" presentation means -> notable-top-5% and "spotlight" means -> notable-top-25%. As stated in our emails, we are disassociating presentation type from AC recommendations

**Summary Of Ac-Reviewer Meeting:**

We entered the meeting with 3 of 4 reviewers voting for at least weak acceptance of the paper, while reviewer 6iPj maintained strong concerns against acceptance.  The main remaining criticisms being:

(a) The theoretical results can not support the safe semi-supervised learning claim. For this point, I think "Debiased SSL" sounds ok.
(b) In the experiments, the authors still use MNIST dataset and the labels for CIFAR-10 dataset (4000) are too many. So the experimental results did not convince me.

Other reviewers shared their feedback around these main points.

Reviewer DVba:
(a) I agree that the authors should try to better clarify / more precisely define their contribution. Regarding the exact choice of terms, the authors did mention in their response the term "unbiased SSL" has been used in the literature to refer to something else. So maybe its ok for authors to keep the  "Safe SSL" in name, but be more clear on what specific subtype of "Safe SSL" they are pursuing in this paper.
(b) Based on the standard SSL evaluation protocol by Oliver et al (2019) (which FixMatch and MixMatch adopts), using 4000 labels for CIFAR10 seems to be a core setting for SSL evaluation. Indeed, the experiment evaluation of this paper can be improved by evaluating more challenging settings (e.g., CIFAR with 40 / 250 labels, as in Table 2 of FixMatch). However, given the theoretical contribution of this paper, I believe the evaluation can be considered as borderline sufficient (hence my score of 6).

Reviewer NKdq:
(a) I agree with [DVba]. Maybe, the authors can clarify what exactly they mean by safe SSL and keep the name given that the unbiased SSL term is used for something else. This point seems minor to me.
(b) I also agree with [DVba] on this point. I agree that the experimental evaluation could be more thorough but given that the theoretical contribution of this work is interesting, it looks fine to me.

Reviewer 1cgk:
(a) I agree with point one as I suggested earlier avoiding the “safe SSL” term.
(b)  Regarding CIFAR-10 with 4K labels, I am ok given the fact that they have conducted experiments on some other datasets.
Considering above issues I still I think  this work has some values (and therefore keep with my rating as before) mainly because of the following reasons:
- Although not true in terms of accuracy, some other reported metrics such as cross-entropy and expected calibration look promising (mainly referring to results in Table 1, 2,3  and 4(per class accuracy)).
- Following my initial review the authors performed some statistical tests and added them in their revised manuscript (sec 4.2, and appendix N).
- Theoretical contribution as some other reviewers have also agreed on


**Outcome:**
Based on this discussion, reviewer 6iPj has increased their score. In my meta-review I also indicate to the authors that clarification around the term "safe SSL" is needed in the final iteration of the paper.